# Association of Ultrasonography and Health Education during Antenatal Visits among Pregnant Women to Reduce Unnecessary Caesarean Section in Bangladesh: Study Protocol for a Cluster Randomized Control Trial

**DOI:** 10.3390/mps5060101

**Published:** 2022-12-17

**Authors:** Habiba Shirin, Michiko Moriyama, K.A.T.M. Ehsanul Huq, Md Moshiur Rahman, Sumaya Binte Masud, Rowshan Ara Begum, Kamrunnahar Misty, Mohammad Delwer Hossain Hawlader

**Affiliations:** 1Graduate School of Biomedical and Health Sciences, Hiroshima University, Hiroshima 734-8553, Japan; 2Obstetrical and Gynaecological Society of Bangladesh (OGSB), Dhaka 1207, Bangladesh; 3Department of Public Health, North South University, Dhaka 1229, Bangladesh

**Keywords:** unnecessary, caesarean section, ultrasonography, health education, Bangladesh

## Abstract

The rate of caesarean section (C/S) has been increasing globally. The proposed study aims to evaluate the effects of ultrasonography (USG) and health education in reducing unnecessary C/S among pregnant women. The secondary objectives are to increase antenatal care (ANC) and post-natal care (PNC) visit rates, increase USG use, increase institutional (hospitals and clinics) delivery, reduce delivery related complications and reduce the number of still births. This is a prospective, open-label, two-arm cluster randomized controlled trial (RCT). A total of 288 pregnant women are enrolled from two urban and two rural hospitals in Bangladesh. Women were screened during their first ANC visits, then written informed consent was taken from the participants. Women in intervention hospitals receive eight ANC visits, two additional USG, and health education eight times during their ANC visits. In contrast, participants at control hospitals receive usual care consisting of four ANC visits and two USG during their ANC visits. The primary outcome is to compare the rate of unnecessary C/S during PNC visits. This study obtained approval from the Institutional Review Board of North South University (2021/OR-NSU/IRB/0804), Bangladesh in November 2021 and was registered in clinicalTrails.gov **(#**NCT05135026).

## 1. Introduction

The World Health Organization (WHO) considered that the standard rate for caesarean section (C/S) should be between 10.0% and 15.0% [1]. However, since 2000 the rate of C/S has been increasing globally, by about twofold from 12.1% to 21.0% in 2021. The trend is increasing and it is predicted that C/S will reach nearly 29.0% in 2030 [2]. The highest rate was observed in Latin America and Caribbean countries (43.0%) and, in contrast, the lowest rate was observed in the countries of sub-Saharan Africa (5.0%) in 2021 [2]. Concurrently with C/S, labor induction increased and instrumental delivery decreased over this time. Indication for C/S as urgent necessity was due to prolonged labor followed by impending fetal asphyxia. Unfortunately, most of the planned C/S were conducted at the mother’s request due to fear of vaginal delivery, followed by fear of increased previous uterine scars [3].

In Bangladesh, C/S continued to increase from 4.0% to 31.0% between 2004 and 2016 and reached almost 33.0% in 2017–2018 [4]. Medically unnecessary C/S was estimated to be about 77% of all C/S in 2018 and increased from 66% since 2016 [5]. The economic burden of each C/S is on average USD 612 and, unfortunately, each patient spends this amount of money out of pocket [5]. It was found that C/S was almost five times more predominant among the richest compared to poorest quintiles between low- and middle-income countries. The number of deliveries in private facilities executed by C/S was 1.6 times higher compared to deliveries in public facilities [2].

Factors related to C/S are obesity, multiple pregnancies, nulliparous, older age and the first birth (40%) [6]. The other risk factors include age at first birth of the mother, having a wealthy family, higher education, living in urban areas, having lower number of children, large size of the baby, and being a housewife as a predisposing factor [7]. Mother’s willingness, physician factor, increased fear of medical litigation, economic and social factors are also involved in unnecessary C/S [6]. C/S has short-term (anesthesia complication, organ injury, infection, thromboembolic disease for mother and neonatal respiratory distress for baby) and long-term complications (uterine rupture, placenta accrete, placenta previa, ectopic pregnancy, infertility, hysterectomy, intra-abdominal adhesions and subsequent C/S delivery for mother, and asthma and obesity in children), which can last for several years [6].

In 2016, the WHO recommended increasing antenatal care (ANC) from four to eight visits with health care providers throughout the pregnancy period. The recommended first visit is at the first 12 weeks of pregnancy, followed by 20, 26, 30, 34, 36, 38 and 40 weeks. This increased number of ANC visits facilitates the reduction of perinatal death by 8 per 1000 live births, in contrast to the previously set four visits [8]. In a Cochrane review, it was revealed that reduced standard visits increased perinatal deaths and women were less satisfied due to long gaps between the visits [9]. Moreover, this standard care of ANC visits also reduces maternal mortality. The number of antenatal contacts in health care facilities can improve safety during pregnancy by detecting any abnormalities by mother and fetal assessment. More ANC with knowledgeable, respectful and supportive health practitioners influences the positive experience and improves care during pregnancy [10].

In Bangladesh, 81.9% of pregnant women went to at least one ANC visit and 47.0% went for four visits or more in 2017; however, the goal is to achieve 50% by 2022 [4]. Pregnant mothers in urban areas receive (59%) higher than four or more ANC visits compared to rural areas (43%) [4].

A study conducted in Bangladesh observed that 33% of pregnant women were visited in the ANC first contact at 27 to 32 weeks of gestation. Ultrasonogram (USG) facilities were not available in ANC community clinics and therefore, there was low compliance (<50%) for the USG. Moreover, USG facilities were not available in ANC community clinics [11]. It is recommended to perform one USG before 24 weeks of pregnancy to estimate gestational age, detect any fetal anomalies or multiple pregnancies and improve mother’s pregnancy experience. However, if USG is not performed before 24 weeks, a late USG could be considered for multiple pregnancies, presentation and location of the placenta. It was acknowledged that the use of USG before 24 weeks has no effect on reducing perinatal mortality [12] and the use of at least four ANC visits could be increased in terms of institutional delivery [13,14].

As for deliveries, 50% occurred at home and the remaining 50% took place at health care facilities (institutional delivery) and, among these deliveries, in public facilities were 14%, private facilities 32% and in non-government organization (NGO) facilities 4% in 2017–2018 [4]. Home delivery in Bangladesh is performed predominantly by unskilled birth attendants. Mothers who deliver at home receive less postnatal care (PNC) than mothers who deliver at health care facilities (97% vs. 7%) [4]. Therefore, hospital/facility-based delivery is promoted instead of home-based delivery due to shortage of skilled birth attendants, particularly in rural areas. To manage maternal complications and prevent maternal deaths, institutional delivery is essential in Bangladesh [15].

Childbirth and relaxation training, and psychoeducation for pregnant women may reduce unnecessary C/S. [16]. Access to health care facilities, mothers’ education and improved economic conditions could reduce maternal mortality [17]. The maternal mortality ratio (MMR) in Bangladesh decreased from 434/100,000 to 173/100,000 between 2000 and 2017 [18], and neonatal and under-five years old children mortality rates were 23 and 38 per 1000 live births, respectively, in 2017 [19]. However, enormous efforts are still needed to achieve the sustainable development goal 3 (SDGs); the target is less than 70 maternal death per 100,000 live births, and neonatal and under-five mortality rates below 12 and 25 per 1000 live births by 2030 [19,20].

The proposed study aims to evaluate the effects of additional USG and health education in reducing unnecessary C/S among pregnant women. The main hypothesis is that pregnant women who receive additional USG and health education intervention will have a reduced percentage of unnecessary C/S compared with those who receive usual care as a control group. The secondary objectives are to increase ANC and PNC visit rates, to increase USG use, to increase institutional (hospital and clinics) delivery, to reduce the delivery-related complications (ante partum and postpartum hemorrhage (APH, PPH))., and to reduce the number of still births among women in the intervention group.

## 2. Materials and Methods

### 2.1. Trial Design

This is a prospective, open-label, two-arm cluster randomized controlled trial (RCT). The protocol is prepared in accordance with the Standard Protocol Items Recommendations for Interventional Trials [21] and designed and reported in accordance with the Consolidated Standards of Reporting Trials (CONSORT) [22]. (Figure 1). The trial participant enrollment was started on 28 November 2021 and it is expected that the last participant out will be completed by December 2022.

### 2.2. Study Setting

The proposed study is conducted in two urban hospitals (Dhaka Medical College & Hospital (DMCH) and Sir Salimullah Medical College & Hospital (SSMCH)). and two rural hospitals (Bogra 250 bed Mohammad Ali District Hospital (BDH) and General Hospital, Munshigonj (MGH). in Bangladesh.

### 2.3. Study Participants

In the study area, recommended ANC visits are performed at least four times during the duration of pregnancy, the First visit at 8–12 weeks, Second visit at 24–26 weeks, Third visit at 34 weeks and Fourth visit at 36–38 weeks. In the intervention hospitals, we perform 4 additional ANC visits at 20, 30, 36 and 40 weeks (total 8 visits). Patients also receive USG 2 additional times during the Third visit at 24–26 weeks and Fifth visit at 34 weeks (2 routine USG + 2 USG in Third and Fifth ANC visits and even more USG if needed + health education: a pictorial flip chart showing danger signs during pregnancy and potential risks for unnecessary caesarean delivery, in order to increase awareness for safe delivery). In the control hospitals, pregnant mothers receive their routine 2 USG during their usual ANC check-up visits at First visit of 8–12 weeks and Fourth visit of 38 weeks (Figure 2). In the intervention and control hospitals, we collected socio-demographic factors, clinical characteristics and pregnancy outcomes during ANC and PNC visits from the pregnant mothers.

### 2.4. Inclusion Criteria

All the pregnant mothers irrespective of their age who attended the designated study hospitals; all pregnant mothers who have/do not have complications regarding a delivery outcome with an indication of normal delivery or C/S; all pregnant mothers who are willing to participate in the study. Exclusion criteria included: Not willing to participate; early pregnancy with an indication for C/S (co-morbidities, H/O previous C/S, etc.)

### 2.5. Operational Definition of Term

Unnecessary C/S: when women request from the physician a delivery without any acceptable indication of C/S [23].

Planned C/S or elective caesarean delivery: Medical indication may be present or not. Primary indication for planned C/S such as malpresentation, narrow pelvis, previous uterine scar, or twin pregnancies with the first twin in a breech presentation. If there is no medical indication, then planned C/S is also called unnecessary C/S [23].

Emergency C/S: Needs to perform C/S very quickly with a proper indication due to the immediate health concern of mother and/or baby [23].

### 2.6. Recruitment Procedure, Obtaining Consent and Checking Eligibility

Participants were enrolled in a hospital-based intervention group (*n* = 144) and a control group (*n* = 144). In the intervention and control hospitals, pregnant mothers usually visit the outpatient antenatal corner for their ANC check-ups. They are identified during that visit by midwives. If any had missed a menstrual period for more than 1.5 months, then midwives perform a pregnancy test using a pregnancy test kit. If a woman is diagnosed as pregnant, then, midwives provide a ‘Health Service Card’ with a schedule of ANC visits. They also provide iron and folic acid tablets as soon as pregnancy is diagnosed. The women are advised to start taking one tablet daily from the next day throughout the pregnancy period. At the time of the first ANC, midwives check the inclusion and exclusion criteria for eligibility to enroll into the study. After midwives have written informed consent, they perform a face-to-face interview and fill out the questionnaires. They perform a clinical examination, take gynecological and obstetrics history, anthropometric measurements including height, weight and mid upper arm circumference (MUAC) and laboratory, and data are entered into a secured computer.

The pregnant mothers in the control and intervention groups undergo the following procedure during their ANC visits:

Control Group (CG):

In the control hospitals, participants receive usual care (recommended at least 4 ANC and 1 PNC visits with 2 times USG). They do not receive health education.

1st visit (8–12 weeks): They receive a physical examination, height, weight, blood test for hemoglobin (Hb), blood grouping and Rh typing, random blood sugar (RBS) and USG for the 1st time. They undergo routine urine and microscopic examination (R/M/E) if they have any complaint of burning sensation during micturition for suspected urinary tract infection (UTI).

2nd visit (24–26 weeks): They receive physical examinations to check for any obstetric complications. They start to receive calcium tablets during this visit.

3rd visit (34 weeks): They receive a physical examination and are checked for any obstetric complication; such as vaginal bleeding, insufficient fetal movement and lower abdominal pain.

4th (last) visit (38 weeks): They receive a physical examination and check for any obstetric complication. If there are any suspected cases of anemia, then they receive blood test for Hb. They undergo routine USG for the 2nd time and are given advice for safe delivery.

5th (PNC) visit (post 4–6 weeks): After the delivery (4–6 weeks) mother and infant will receive a post-natal check-up at the hospitals. Midwives take information about the delivery history and pregnancy outcome.

Intervention Group (IG):

According to the proposed trial design shown in Figure 2, when participants come for their 1st ANC visit, they are asked to ensure 8 ANC visits and 1 PNC visit with 4 times USG, even more if needed. Additionally, they receive health education 8 times in their 8 ANC visits. Therefore, they receive 4 more ANC visits than recommended, 2 additional USG in the 3rd and 5th ANC visits and health education 8 times.

1st visit (8–12 weeks): The same procedure as the CG with the addition of health education.

2nd visit (20 weeks): Physical examinations to check for any obstetric complication. Health education.

3rd visit (24–26 weeks): Physical examination to check for any obstetric complication. Calcium tablet. USG for the 2nd time and health education.

4th visit (30 weeks): Physical examination and check fir any obstetric complication, such as vaginal bleeding, insufficient fetal movement, and lower abdominal pain. Health education.

5th visit (34 weeks): Physical examination and check for any obstetric complication, such as bleeding, insufficient fetal movement and lower abdominal pain. USG for the 3rd time and health education.

6th visit (36 weeks): Physical examinations and check for obstetric complications, such as vaginal bleeding, insufficient fetal movement and lower abdominal pain. Health education.

7th visit (38 weeks): Physical examinations and check for any obstetric complications. If there are any suspected case of anemia, then they receive blood test for Hb. USG for the 4th time, health education and advice for safe delivery.

8th visit (40 weeks): Physical examination and check for any obstetric complication. Health education and advice for safe delivery.

9th (PNC) visit (post 4–6 weeks): After the delivery (4–6 weeks), mother and infant receive a post-natal check-up at the hospital. Midwives take information about the delivery history and pregnancy outcome.

In DMCH, SSMCH and two rural hospitals, there are facilities for USG performed by a trained sonologist. After conducting USG, they provide reports with printouts of USG images for the pregnant mothers. If there are any abnormalities, they are advised to consult with a gynecologist from the respective hospital.

### 2.7. Allocation and Randomization

This is a multicenter, prospective, two-arm (1:1), open label cluster randomized controlled study. To minimize bias, we used block randomization. We divided the hospitals into urban and rural, then from the urban area (DMCH and SSMCH), we randomly selected one hospital for the IG and another hospital for the CG and, from the rural area (BDH and MGH), one hospital was randomly selected as intervention site and the other hospital as comparison site. One member of research staff, who was not involved in any of the research activity in this trial, performed this computer-generated randomization.

### 2.8. Health Education Materials

In each antenatal care visit, pregnant mothers in the intervention group received health education during their pregnancy period to increase their knowledge and hence make decisions for better pregnancy outcome. Health education is provided face-to-face by midwives in the local (Bengali) language. We developed health education materials on the basis of the study outcomes. At first, we checked the existing education materials used for pregnant mothers in the ANC clinics and hospitals. We combined this information with our outcomes related to pregnancy and delivery. In the education module, we used colourful pictures and charts in the native language for easy understanding. The education material contains information about healthy behaviour, health care utilization pattern, pregnancy & delivery related complications and consequences. It contains pregnancy related information about regular intake of nutritious food with additional calories, benefits of healthy food, problems from poor nutrition, personal hygiene; especially hand washing, adequate rest and sleep, intake of iron pills and folate tablets, proper immunization, benefits of exclusive breast feeding, saving money for hospital, transport and delivery cost, arranging vehicles for shipment from home to hospital for delivery. The education materials also contained information on advantages of regular antenatal care visits, danger signs during pregnancy, advantages of ultrasonography, importance of normal delivery, risk and indication (when needed) of caesarean section, hospital delivery and still birth, and benefits of post-natal care for mother and baby. In the education material, we used a pictorial flip chart showing danger signs during pregnancy and potential risks for unnecessary caesarean delivery to increase awareness of safe delivery.

### 2.9. Quality Control and Quality Assurance for Data Management

All the research investigators periodically monitor and assess the data quality and timeliness, participant recruitment, accrual and retention, participant risk versus benefit, performance of trial sites and other factors that can affect the study outcome. An independent Data Safety and Monitoring Board (DSMB) was formed by the Institutional Review Board (IRB) of North South University (NSU) for appropriate oversight and monitoring of the conduct of clinical trials to ensure the safety of participants and the validity and integrity of the data. The DSMB is comprised of two members who are not involved in conducting this study. Every six months, they call a meeting to discuss study progress, including any ethical issues such as deviations from the study plan, misconduct, or serious adverse events. The DSMB may invite the PI or designee to provide information on study conduct, present data, or respond to questions at any time. They may request any data for assessment and may review unblinded data at any time.

### 2.10. Training of Study Staff

All the research investigators, clinical and laboratory staff who are directly involved with the study participants took the Research Ethics Online Training course supported by WHO for protecting research participants before initiation of the study. Research investigators and midwives received protocol specific training from the PI and other investigators in different sessions. They receive monthly refresher training on how to take informed consent, fill questionnaires, provide health education and take anthropometric measurements.

### 2.11. Data Collection and Entry Procedures

At the time of the interview, midwives collect sociodemographic information (i.e., age, occupation, etc.), gynecological and obstetrics history (i.e., abortion, stillbirth, gestational diabetes, etc.), clinical examination (i.e., BMI, MUAC, blood pressure (BP)), laboratory investigation (i.e., blood, USG, urine) and pregnancy outcomes (i.e., normal delivery or C/S).

Questionnaires are sent from the hospitals to the NSU research room on a weekly basis. After receiving the questionnaires, the study Research Officer enters the data into a secured computer and this is de-identified before analysis by using SPSS for Windows (Version 25.0; SPSS Inc, Chicago, IL, USA).

### 2.12. Sample Size

We assumed 33% (the national prevalence of C/S in Bangladesh) [4] exposure rates of C/S for IG and CG. At a 5% level of significance with 80% power, we estimated the sample size of 274–137 each in CG and IG (case and control ratio of 1:1 and desired odds ratio 2.0). With the presumption that 5% of participants would be withdrawn and lost to follow up (7 subjects) during the study procedures, the estimated sample size was 144 in each group and 144 × 2 = 288 in total [24].

### 2.13. Statistical Analyses

An intention-to-treat analysis is part of the study [25]. Frequencies and percentages of different demographic and clinical baseline characteristics will be described and compared between the participants at intervention and control hospitals. Mean and standard deviation (SD) or median (minimum-maximum) will be expressed for continuous variables. Differences in proportions for the categorical variables will be compared by chi-square test. For the data, which is normally distributed, differences in means of continuous variables will be compared by Student’s t-test, and for comparing data that may not be normally distributed, the Mann-Whitney U test will be used. A probability of less than 0.05 will be considered as statistically significant. The strength of association will be determined by calculating odds ratio (OR) and its 95% confidence intervals (CI) by adjusting all the possible confounding factors. We will use these statistics both in bivariate analyses and multiple logistic regressions.

### 2.14. Strengths and Limitations of This Study

➢As this is a multicenter study, it will give us a broader understanding of the risk factors related to unnecessary C/S and will minimize biases.➢For resource poor low-and-middle income countries, this study’s findings will give significant economic relief to pregnant mothers by preventing unnecessary C/S and reduce hospital costs and infections.➢The study findings will have a significant impact in reducing maternal and childhood morbidity and mortality and increase the potential of achieving the Sustainable Development Goal (SDGs) by 2030.➢According to the study design, pregnant women maybe admitted to any of the hospitals according to their choice. As there are many public-private hospitals and clinics in the study areas, we could not involve all the gynecologists and obstetricians from the study area in our study. According to World Health Organization, there are roles played by gynecologist and obstetrician in C/S, and the outcome might be influenced by these in terms of their decisions to perform C/S.

## Figures and Tables

**Figure 1 mps-05-00101-f001:**
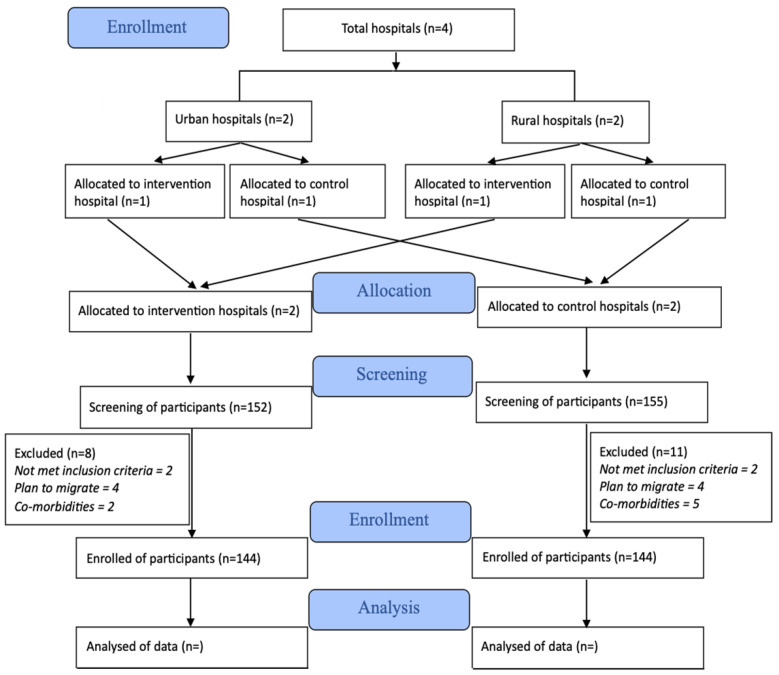
CONSORT Flow Chart.

**Figure 2 mps-05-00101-f002:**
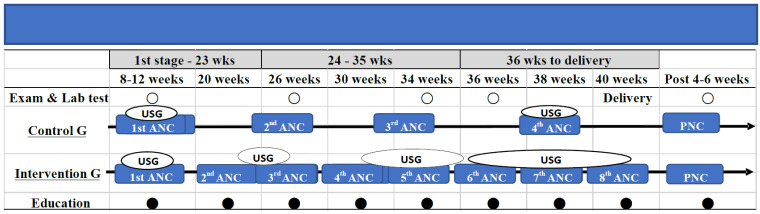
Activities compared Control and Intervention group.

## Data Availability

Not applicable.

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
