# Peer review of "Association of Ultrasonography and Health Education during Antenatal Visits among Pregnant Women to Reduce Unnecessary Caesarean Section in Bangladesh: Study Protocol for a Cluster Randomized Control Trial"

_mps, 2022, doi:10.3390/mps5060101_

Round 1
Author Response
Date: December 07, 2022
Subject: Re-submission of the Manuscript ID: mps-2020865
Thank you very much again for your valuable comments and further consideration of my manuscript for possible acceptance.
Comment: Minor revisions (typos): Fig 1: middle column: Enrollment
Response: We corrected accordingly within the revised manuscript.
Best regards,
Habiba Shirin, MBBS, DMU, MHS
Graduate School of Biomedical and Health Sciences
Hiroshima University, Japan
Mobile:080 2018 6606

Reviewer 2 Report
Congratulation on working in such an essential field of obstetrics!
It is a critical topic, and I highly congratulate the author's effort.
But – I see that a study will be concluded next month and I do not see any reason to publish a protocol that we can not influence and give some constructive suggestions.
I suggest the authors conclude the research, analyze the results as soon as possible, and with all these data publish the article. They have the introduction and methods part written yet.
I suggest thorough English language editing and typo and missing letters and missing word corrections!
Author Response
Date: December 07, 2022
Subject: Re-submission of the Manuscript ID: mps-2020865
Thank you very much again for your valuable comments and further consideration of my manuscript for possible acceptance.
Best regards,
Habiba Shirin, MBBS, DMU, MHS
Graduate School of Biomedical and Health Sciences
Hiroshima University, Japan
Mobile:080 2018 6606
Comments: But – I see that a study will be concluded next month and I do not see any reason to publish a protocol that we can not influence and give some constructive suggestions.
I suggest the authors conclude the research, analyze the results as soon as possible, and with all these data publish the article. They have the introduction and methods part written yet.
Responses: Thank you very much for your valuable suggestion. According to our primary and secondary objectives, we planned to publish several papers based on the individual objective. That is the requirement for completion of my PhD graduation. In this study protocol paper, we have all the primary and secondary objectives and we can disseminate the knowledge of problems related to mother and child health and possible solutions by conducting this research.
Comment: I suggest thorough English language editing and typo and missing letters and missing word corrections!
Response: To improve the English revision, all the authors worked on it and also checked by a native English language person from Hiroshima University Writing Center, 1-2-2 Kagamiyama, Higashi-Hiroshima City Hiroshima, Japan 739-8512, TEL: +81-82-424-6201.

Reviewer 3 Report
Dear authors,
I really enjoyed your manuscript and I have some minor comments to it:
1. The English needs improvement. I suggest you ask an English speaker to review it before re-submitting it.
2. The study is an on-going study so you should use the present tense throughout your manuscript (not the future, as you have used it). Please modify the tense of the verbs where appropriate.
3. Some information are missing, as highlighted in the text.
Great job!

Author Response
Date: December 07, 2022
Subject: Re-submission of the Manuscript ID: mps-2020865
Thank you very much again for your valuable comments and further consideration of my manuscript for possible acceptance.
Best regards
Habiba Shirin, MBBS, DMU, MHS
Graduate School of Biomedical and Health Sciences
Hiroshima University, Japan
Mobile:080 2018 6606
Comment 1. The English needs improvement. I suggest you ask an English speaker to review it before re-submitting it.
Response: To improve the English revision, all the authors worked on it and also checked by a native English language person from Hiroshima University Writing Center, 1-2-2 Kagamiyama, Higashi-Hiroshima City Hiroshima, Japan 739-8512, TEL: +81-82-424-6201.
Comment 2. The study is an on-going study so you should use the present tense throughout your manuscript (not the future, as you have used it). Please modify the tense of the verbs where appropriate.
Response: Thank you very much for your valuable suggestion. We changed all the future tenses where appropriate throughout the manuscript.
Comment 3. Some information are missing, as highlighted in the text.
Response: According to your suggestion, we changed all in the text.

Round 2
Reviewer 2 Report
I thank you for your efforts and hope you manage to steer things in the right direction in your country!